# Spin-controlled atom–ion chemistry

Tomas Sikorsky[1], Ziv Meir[1], Ruti Ben-shlomi[1], Nitzan Akerman[1] & Roee Ozeri[1]

Quantum control of chemical reactions is an important goal in chemistry and physics. Ultracold chemical reactions are often controlled by preparing the reactants in specific quantum states. Here we demonstrate spin-controlled atom–ion inelastic (spin-exchange) processes and chemical (charge-exchange) reactions in an ultracold Rb-Sr$^+$ mixture. The ion's spin state is controlled by the atomic hyperfine spin state via spin-exchange collisions, which polarize the ion's spin parallel to the atomic spin. We achieve ~ 90% spin polarization due to the absence of strong spin-relaxation channel. Charge-exchange collisions involving electron transfer are only allowed for (RbSr)$^+$ colliding in the singlet manifold. Initializing the atoms in various spin states affects the overlap of the collision wave function with the singlet molecular manifold and therefore also the reaction rate. Our observations agree with theoretical predictions.

[1] Department of Physics of Complex Systems, Weizmann Institute of Science, Rehovot 7610001, Israel. Correspondence and requests for materials should be addressed to T.S. (email: tomas.sikorsky@weizmann.ac.il)

The control of ultracold collisions between neutral atoms is an extensive and successful field of study. The tools developed in this field allow for ultracold chemical reactions to be managed using magnetic fields[1], light fields[2], and spin-state manipulation of the colliding particles[3] among other methods. Control of chemical reactions in ultracold atom–ion collisions is a young and growing field of research. Recently, the collision energy[4] and the ion electronic state[5–8] were used to control atom–ion interactions.

Ultracold collisions are an important tool for manipulating atomic gases. The cross-section for elastic collisions and inelastic reactions typically depends on the combined spin-state of the colliding atoms. The rate of inelastic processes can therefore be controlled by the atomic spin. Remarkable examples include molecular association and three-body recombination close to a magnetic Feshbach resonance[9,10]. Ultracold atom–ion collisions are studied in several laboratories and efforts to gain control over different collisional properties are ongoing[4–8,11–14]. Precise control over ultracold atom–ion collisions has rich prospects such as emulating solid-state systems[15], performing atom–ion entanglement[16], quantum gates[17], and the formation of mesoscopic ions[18]. The research of ultracold atom–ion collisions can also lead to better understanding of interstellar molecular formation[19]. In recent experiments, different inelastic collision rates were shown to depend on the collision energy as well as the electronic state of an atom–ion system[5–8]. However, no spin control of different collisional properties was demonstrated to date.

Although the spin of both ultracold atoms and ions can be prepared in a precise predetermined state, for this initial spin state to control a collisional process, the total spin of the system has to be conserved during the collision. Thus, spin dynamics during the collision has to be dominated by spin exchange and the relaxation of spin through, e.g., coupling to orbit has to be negligible. In ultracold atomic gases, the absence of spin relaxation in collisions has enabled the magnetic trapping of atoms[20] and has led to the realization of a (SWAP)$^{1/2}$ gate[21]. Spin exchange induced spin locking, and collective spin excitation were observed in BEC[22] and non-degenerate gases[23]. Spin-exchange collisions between noble gases and alkali atoms were used to polarize the nuclear spin of the noble-gas atoms[24]. The only study, so far, of spin dynamics in ultracold atom–ion systems was performed in a Yb$^+$-Rb mixture where it was found that it is dominated by spin relaxation due to second-order spin orbit coupling[25,26].

Here we report the study of atom–ion collisions in an ultracold spin polarized mixture of Sr$^+$-Rb. We find that spin dynamics during a collision is dominated by spin exchange and spin relaxation is largely suppressed. By preparing the atoms in different initial spin states, we demonstrate control over two inelastic collision rates. First, we can turn spin exchange off and on by preparing the ion spin parallel or antiparallel to that of the surrounding atomic cloud. As a consequence, by immersing an unpolarized single ion in a spin-polarized atomic bath, we observe that the ion spin is polarized through collisions. Second, we study the rate of charge-exchange reactions of the polarized atom–ion mixture. The radiative decay into a singlet ground-state of Rb$^+$-Sr can only proceed from the singlet state of our entrance channel (see Fig. 1c). We control the charge-exchange rate by controlling the projection into singlet and triplet state.

## Results

**Spin polarizing the Sr$^+$ ion with ultracold atoms.** In our experiment, a single spin-polarized $^{88}$Sr$^+$ ion is trapped in a linear Paul trap, ground state cooled to ~40 μK, and then immersed into an ultra-cold (~3 μK) and hyperfine spin-polarized $^{87}$Rb cloud trapped in an optical dipole trap[27]. Owing to non-equilibrium dynamics of atom–ion elastic collisions, the ion heats to a few mK temperature after several collisions[13]. The Langevin collision rate is 1 kHz. Both species have a single electron in the valence shell. Although $^{87}$Rb has a $I = 3/2$ nuclear spin and a hyperfine-split ground-state manifold, $^{88}$Sr$^+$ has no nuclear spin and a Zeeman split two-fold ground state. The different spin states of both species in the $5S_{1/2}$ ground states are shown in Fig. 1a,b. Following a short interaction time, both the spin of the ion and the density of atoms are measured. See methods section for details. During a collision the two-electron molecular system splits into a triplet, $^3\Sigma^+$, (red dashed line in Fig. 1c) and singlet $^1\Sigma^+$, (blue solid line) spin manifolds, which are energetically

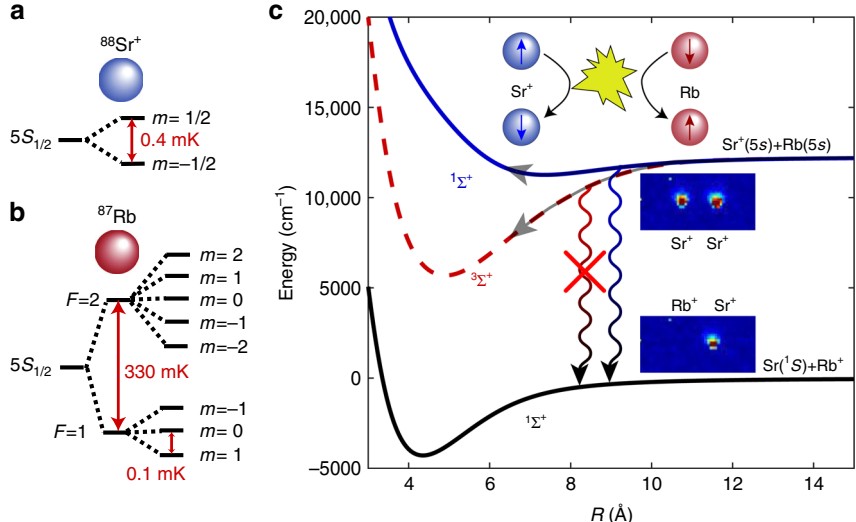

**Fig. 1** Energy levels diagrams. **a**, **b** Level structure of the $^{88}$Sr$^+$ electronic ground state and hyperfine structure of the $^{87}$Rb. The Zeeman splitting is for $B = 3$ G. **c** Pictorial representation of potential energy curves of the (RbSr)$^+$ complex. The experimental entrance channel (Sr$^+$(5s)+Rb(5s)) is not the absolute ground state of the system which allows for radiative charge-exchange processes (curly lines). During a collision, the atomic asymptotic state (Sr$^+$(5s)+Rb (5s)) splits into a superposition of singlet ($^1\Sigma^+$, blue solid line) and triplet ($^3\Sigma^+$, red dashed line) states. Only radiative charge exchange from the singlet state is allowed (blue curly line), as the molecular ground state of the system (Sr($^1$S)+Rb$^+$) is also a singlet state ($^1\Sigma^+$, solid black line). A pictorial representation of spin-exchange collision is also shown

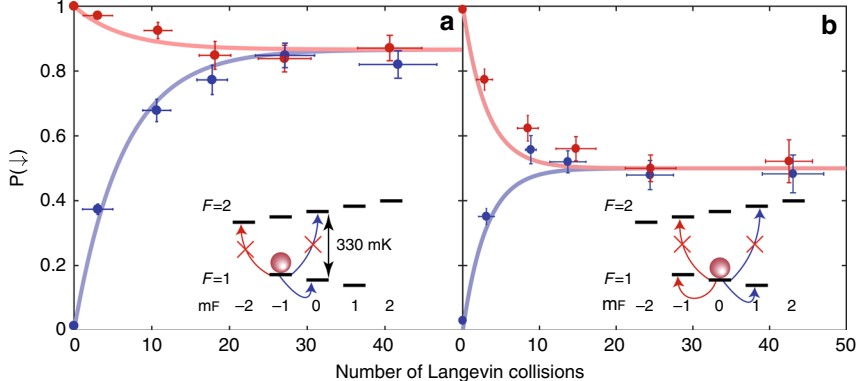

**Fig. 2** Collisional pumping of the spin of the ion. $Sr^+$ ion spin projection on the $|\downarrow\rangle_{Sr^+}$ state, $P(\downarrow)$, as function of number of Langevin collisions (time). In blue (red) the ion is prepared in the $|\uparrow\rangle_{Sr^+}$ $(|\downarrow\rangle_{Sr^+})$ spin-state. The atoms are prepared in state $|1,-1\rangle_{Rb}$ (**a**) or $|1,0\rangle_{Rb}$ (**b**). Insets show energetically allowed and forbidden spin-exchange processes. Error bars represent 1 SD

separated due to the Pauli exclusion principle and the Coulomb interaction.

The spin-exchange interaction conserves the total two-electron spin projection along any direction. Thus, under spin exchange, if the atom and ion are prepared with parallel electronic spins, they collide on the triplet, $^3\Sigma^+$, molecular potential and their spin states do not change. However, when initialized with anti-parallel electronic spin states, the atomic states are split into a superposition of singlet and triplet manifolds during the collision. The singlet and triplet wavefunctions acquire different phases which results in a finite probability for the spin states to be exchanged[28],

$$\sigma_{\text{exch}} = \left| \langle \Psi_{\text{init}} | \hat{\mathbf{S}}^{(\text{Rb})} \cdot \hat{\mathbf{S}}^{(\text{Sr}^+)} | \Psi_{\text{final}} \rangle \right|^2 \cdot \frac{4\pi}{k^2} \sum_{l=0}^{\infty} (2l+1) \sin^2(\phi_{s-t}). \quad (1)$$

Here, $\sigma_{\text{exch}}$ is the cross-section for the spin-exchange process, $\hat{\mathbf{S}}$ is the total electron spin operator, $\phi_{s-t}$ is the phase difference between the singlet and triplet parts of the wavefunction, $k$ is the wave number of relative motion, and $l$ is the relative angular momentum quantum number.

Under spin–orbit interaction, the singlet and triplet states ($^1\Sigma^+$ and $^3\Sigma^+$) are not eigenstates anymore, the total spin projection is no longer conserved, which leads to spin relaxation. To distinguish between spin exchange and relaxation we initialize the ion and atoms with parallel electronic spins. To this end, the ion is prepared in the $|\downarrow\rangle_{Sr^+}$ spin state and the atomic cloud is prepared in the $|2,-2\rangle_{Rb}$ stretched state of the $F=2$ hyperfine level (for a level diagram see Fig. 1). After an interaction time of 500 ms, during which 100's of Langevin collisions occurred, we found that the ion has heated up to a temperature of ~ 20 mK. This heating is likely to be due to the occasional hyperfine energy release owing to spin-relaxation. Furthermore, as this steady-state temperature is much lower than the hyperfine energy gap of 330 mK, the spin-relaxation rate is significantly lower than the elastic Langevin collision rate, which sympathetically cools the ion. As in these temperatures the ion is no longer in the Lamb–Dicke regime, spin detection using electron shelving on a narrow optical transition is no longer reliable. We, therefore, turned to measuring spin dynamics when the atomic cloud is spin-polarized in the $F=1$ ground hyperfine level.

As the collisional energies are on the mK energy scale, spin exchange between $Sr^+$ and Rb prepared in the $F=1$ state is allowed only as long as it does not require Rb to change its hyperfine state and climb the 330 mK hyperfine energy gap. Thus,

when initializing Rb to $|1,-1\rangle_{Rb}$, spin exchange is possible only with $Sr^+$ initialized in the $|\uparrow\rangle_{Sr^+}$ state. Spin exchange with Rb initialized to $|1,0\rangle_{Rb}$ is allowed for both spin directions of $Sr^+$. See Supplementary Eqs. 2, 3, Supplementary Note 1, and the inset of Fig. 2 for detailed information. Figure 2 also shows the measured spin projection on the $|\downarrow\rangle_{Sr^+}$ state, $P(\downarrow)$, as function of number of Langevin collisions for $Sr^+$ prepared in $|\uparrow\rangle_{Sr^+}$ (blue) and $|\downarrow\rangle_{Sr^+}$ (red) and the atomic cloud in (a) $|1,-1\rangle_{Rb}$ or (b) $|1,0\rangle_{Rb}$. As seen, in the case of $|1,0\rangle_{Rb}$, as spin exchange is allowed for both spin states of the $Sr^+$, the ion evolves to a fully mixed spin state. In the $|1,-1\rangle_{Rb}$ case however, spin exchange is largely suppressed when the ion is prepared in $|\downarrow\rangle_{Sr^+}$. Moreover, when the ion is initialized in $|\uparrow\rangle_{Sr^+}$, spin-exchange flips its direction to $|\downarrow\rangle_{Sr^+}$ where it remains. Collisional spin pumping in this case polarizes the ion spin to a steady state of $P(\downarrow) \sim 0.9$. The spin exchange rate can be therefore controlled by manipulating the spin state of Rb.

The steady-state polarization of the ion spin when the atoms are initialized in $|1,-1\rangle_{Rb}$ is limited to $P(\downarrow) \sim 0.9$ due to the spin relaxation. From a fit to a rate-equation solution (see Methods) we found that the spin-exchange rate in our system is $\tau_{SE}/\tau_L = 9.1 \pm 0.59$, whereas the spin-relaxation rate is $\tau_{SR}/\tau_L = 47.5 \pm 6.6$ where $\tau_L = \gamma_L^{-1}$ is the Langevin time constant. The fact that the spin-relaxation rate is ~ 5 times slower than the spin-exchange rate allows us not only to control the spin state of the ion using the atoms but also to maintain the spin state during multiple collisions.

**Spin-controlled charge exchange between the $Sr^+$ and Rb.** We now turn to discuss the effect of spin polarization on reactive collisions. Charge exchange between an alkali atom and an alkali-earth ion is a prototype of a chemical reaction where open-shell reactants exchange an electron and form closed-shell products. Charge exchange in cold atom–ion systems was studied in several experiments[5–8,11,29], but only few were performed without optical mixing of ground and excited states[5,6,29]. Charge exchange, in a heteronuclear atom–ion mixture, can happen in several different ways. First, it can occur as a radiative process where excess energy is carried away or absorbed by a photon. Second, it can happen as a nonradiative process where energy transfers into motional degrees of freedom due to non-adiabatic crossing between molecular potential curves[7]. Finally, at high densities (~ $10^{18}$ m$^{-3}$), non-radiative charge exchange can proceed through three-body recombination where two atoms bind on the charge-exchange potential and energy is carried away by a third atom[30]. In our experiment, due to absence of curve crossings in the entrance channel below the dissociation limit and low atomic densities (~ $10^{17}$ m$^{-3}$), we expect charge exchange to occur radiatively.

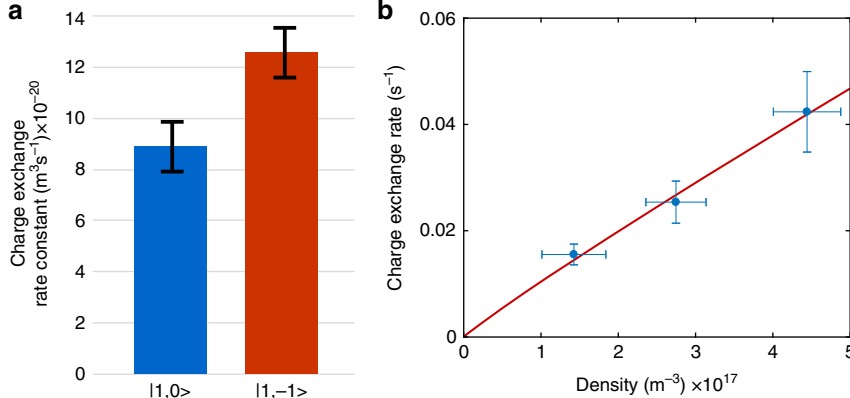

**Fig. 3** Charge-exchange control. **a** Charge-exchange rate for different initial hyperfine states of Rb atoms. **b** Density dependence of the charge-exchange rate averaged for $|1, 0\rangle_{Rb}$ and $|1, -1\rangle_{Rb}$. From a fit to a power-law we estimate the charge-exchange scaling on the density to be $k_{CE} \propto \rho^{0.94(8)}$. Error bars represent 1 SD

As Sr has higher ionization energy than Rb, the entrance channel $Sr^+(5s) + Rb(5s)$ is not the molecular ground state (see Fig. 1). Charge exchange involves both valence electrons moving into the 5s state of the neutral Sr atom, while leaving an ionized Rb without any free electrons. In the absence of a spin–orbit coupling, radiation can only couple the singlet molecular state of $Sr^+(5s) + Rb(5s)$ to the charge-exchanged ground molecular state. As a result, chemical reactions can be triggered or suppressed by initializing the collision in a particular superposition of singlet and triplet states. As the spin state of the ion is driven to a steady-state polarization by the atomic bath, control of the atomic spin determines the reaction rate. Unlike previous experiments where the charge-exchange rate was modified by initializing atoms in different excited states[5,6], here both atoms and ion are in the ground electronic state.

Although, in our experiment, we observe charge-exchange reactions every ~ $5 \times 10^4$ Langevin collisions when Rb is prepared in the $F = 1$ hyperfine level, we do not observe charge-exchange reactions when it is initialized in $F = 2$. A similar suppression was previously reported in a $Yb^+$-Rb mixture[5], where the suppression was attributed to the difference in hyperfine interaction. An alternative explanation, in our case, would be a suppression of charge exchange due to the increase in steady-state temperature of the ion when Rb is initialized in $F = 2$ and the hyperfine energy is occasionally released. Preliminary investigations have shown that comparable suppression occurs when Rb is initialized in $F = 1$ and the ion is heated to similar temperatures using excess micromotion (see Supplementary Note 2 and Supplementary Figure 1). An investigation of the dependence of charge exchange on the reaction energy is underway.

In our experiment, charge-exchange events were identified by the disappearance of ion fluorescence. To corroborate that these events are indeed charge-exchange events, we performed a similar experiment using a two-ion crystal. We verified that every time ion fluorescence disappeared, a dark ion remained in the crystal and used resonant excitation mass spectroscopy[31] to determine the mass of the reaction product, which consistently indicated $Rb^+$ (see the inset of Fig. 1). Furthermore, to verify that in our experiment charge exchange is a two-body process, which supports a radiative mechanism, we measured the charge-exchange rate at different densities and recovered a linear density dependence; see Fig. 3b. We also performed a charge-exchange experiment at various YAG laser intensities and found no charge-exchange rate dependence on YAG laser power (see Supplementary Note 3 Supplementary Figure 2).

As charge exchange is suppressed when Rb is initialized in the $F = 2$ level, we compared charge-exchange rates when the atoms

are polarized to different spin states in the $F = 1$ manifold. Preparing the atoms in the $|1, -1\rangle_{Rb}$ or $|1, 0\rangle_{Rb}$ states results in different overlap with the singlet state. For atoms prepared in $|1, -1\rangle_{Rb}$ and the ion collisionally spin-pumped to $P(\downarrow) \sim 0.9$, the probability of colliding on the singlet potential curve is 0.3625 (see Supplementary Eq. 1). When the atoms are in a $|1, 0\rangle_{Rb}$ and the ion is in a fully mixed spin state this probability is 0.25. We therefore expect a ratio of 1.45 between the charge-exchange rates in the two cases. We overlap the ion and atoms for 1 s, which is equivalent to ~ $10^3$ Langevin collisions. This was repeated 1000 times in an interlaced manner, in which atoms are prepared alternatively in a $|1, -1\rangle$ and $|1, 0\rangle$ states. During these 2000 repetitions, we recorded 104 charge-exchange events. Sixty-one of these events were recorded when the atoms were prepared in $|1, -1\rangle_{Rb}$ and 43 were with atoms in a $|1, 0\rangle_{Rb}$. This corresponds to a ratio of $1.42 \pm 0.2$ ratio between the rates as expected by the simple considerations above. The measured rates for the two states are shown in Fig. 3a.

## Discussion

In conclusion, here we demonstrate the control of the spin of a single $Sr^+$ ion by spin-exchange collisions with an ultracold bath of Rb atoms. In addition to collisional spin-pumping, we measured a dependence of the charge-exchange reaction rate on the atomic spin and found it to be in good agreement with simple theoretical predictions. Spin control of ultracold atom-ion interactions opens up many exciting possibilities such as the coherent formation of ultracold molecular-ions in their ground state or the study of exotic many-body effects.

## Methods

**State initialization of ultracold atoms and ions.** A more detailed description of the experimental apparatus can be found in a recent publication[27]. We prepare neutral $^{87}$Rb atoms in the specific hyperfine state of the electronic ground state at a temperature of $T \approx 3 \mu K$ in an optical lattice (YAG laser at 1064). We transfer the atoms over 25 cm to the science chamber where they are loaded into a crossed dipole trap ($[\omega_x, \omega_y, \omega_z] = 2\pi \times [0.61, 0.6, 0.1]$ kHz) 50 μm above the $Sr^+$ ion. Here, ~ $10^5$ atoms are spin polarized using a combination of resonant microwave pulses and 780 laser light. The polarization fidelity is above > 99%. The $Sr^+$ ion is trapped in a radiofrequency linear Paul trap with secular trap frequencies of $\omega = 2\pi \times [0.8, 1, 0.4]$ MHz for the two radial and the axial mode, respectively. We performed ground-state cooling and spin-state preparation using a narrow linewidth 674 laser on the $S_{1/2} \rightarrow D_{5/2}$ quadrupole transition. To overlap the atoms with the ion, we move the crossed dipole trap onto the ion position. The experiment was performed at low magnetic field of 3 Gauss; hence, the Zeeman energy splitting has a negligible effect on the energy of the ion.

**State detection of ultracold atoms and ions.** During atom–ion interaction all laser beams are mechanically blocked, except for the off-resonant dipole trap lasers at 1064. After the desired interaction time, we release the atoms from the trap.

After time-of-flight, we detect the number of atoms and their temperature using absorption imaging. The measured density and temperature are used for the atom density estimation. After atomic measurement, we perform Rabi carrier spectroscopy on the narrow $S_{1/2} \rightarrow D_{5/2}$ optical quadrupole transition[27] and Doppler cooling thermometry[32] on the dipole $S_{1/2} \rightarrow P_{3/2}$ transition. We detect charge-exchange events using fluorescence imaging on a charge-coupled device camera.

**Quantitative evaluation of spin dynamics**. We measure the probability of the ion's spin to be in the $S_{1/2}(m = -1/2)$ state ($p_\downarrow$) by shelving $S_{1/2}(m = -1/2) \rightarrow D_{5/2}(m = -5/2)$, and $S_{1/2}(m = 1/2) \rightarrow D_{5/2}(m = 5/2)$ in an interlacing manner. The normalized population is determined by $p_\downarrow = \frac{N_\downarrow}{N_\downarrow + N_\uparrow}$, where $N_\downarrow$ ($N_\uparrow$) are the number of shelving events, indicating the ion is in the $S_{1/2}(m = -1/2)$ ($S_{1/2}(m = 1/2)$) state. The dynamics of a spin in a $|1, -1\rangle_{Rb}$ atomic bath under spin exchange and spin relaxation is governed by a two-level rate equation: $\dot{p}_\downarrow = \gamma_{SE} \cdot p_\uparrow + \gamma_{SR} \cdot (p_\uparrow - p_\downarrow)$ and in a $|1, 0\rangle_{Rb}$ atomic bath by: $\dot{p}_\downarrow = (\gamma_{SE} + \gamma_{SR}) \cdot (p_\uparrow - p_\downarrow)$. $\gamma_{SE}$ ($\gamma_{SR}$) are spin-exchange (spin-relaxation) constants and $p_\uparrow + p_\downarrow = 1$. The collisional rate constant is defined as $k = 1 - e^{-\gamma}$.

**Data availability**. All relevant data are available from the authors on request. We thank Alexei Buchachenko, Timur Tscherbul, Masato Morita, Olivier Dulieu, Edvardas Narevicius and Stefan Wilitsch for helpful discussions. This work was supported by the Crown Photonics Center, ICore-Israeli excellence center circle of light, the Israeli Science Foundation, and the European Research Council (Consolidator Grant No. 616919-Ionology).

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

## Author contributions

T.S., Z.M., and R.O. designed the experiment. T.S., Z.M., R.B., and N.A. performed the experiment and analysed the data. T.S., Z.M., and R.O. wrote the manuscript together.

## Additional information

**Competing interests:** The authors declare no competing interests.

