## [Peer Review File · Nature Communications]

Reviewers' comments:

Reviewer #1 (Remarks to the Author):

The authors investigate inelastic collisions between a spin polarized Sr⁺ ion and spin polarized 87Rb atoms. After a given interaction time they investigate whether a charge exchange has taken place or whether the spin state of the Sr⁺ has flipped. In general, depending on the initial spin states of ion and atoms, the final product populations vary. As a first important feature the authors observe collisional spin pumping--- i.e. collisions with the polarized atoms can polarize the ion spin. In contrast to the work of the Köhl group spin relaxation plays a minor role here and collisions are dominated by spin exchange processes. Another important result is that the data suggest that two-body charge exchange is only taking place within the electronic singlet channel. This is expected if the charge exchange process involves a radiative decay to the electronic ground state.

The presented results are important for the new field of cold atom-ion collisions. The manuscript is written in a clear and intelligible manner. There are, however, a number of typos and some of the English should be checked again.

I recommend the manuscript for publication in Nature Communications after the following issues have been dealt with:

1) On page 2 the authors claim that no spin control of collisional properties was demonstrated to date with cold atom-ion collisions. In view of ref 24 this statement needs to be toned down, in my opinion.

2) page 1: "... can only be reached by initially overlapping a singlet manifold..." This sentence is not clear.

3) Equation (1). The equation seems wrong. The tensor product should be a scalar product.

4) page 2: "Spin-orbit interactions mixes... and leads to spin relaxation." The singlet-triplet mixing is not what is really behind spin relaxation.

5) "Spin-projection is ... no longer conserved". This statement is somewhat sloppy and not clear enough.

6) "... time of 500ms, during which 10's of Langevin collisions occurred," shouldn't it be hundreds of collisions?

7) page 6: What is the spin relaxation rate for $F = 2$?

8) page 6: Defining ultracold collisions to occur when the mK regime is reached, is kind of arbitrary. A better definition for the realm of ultracold collisions is connected to reaching the s-wave regime.

9) page 10: The following sentence is confusing as it does not clearly explain how the experiment is done: "We overlapped... for a duration of 10^6 Langevin collisions and have recorded 104 ... events". How many runs were done? How long (i.e. how many Langevin collisions) was every run?

Typos:

gasses -> gases;

hyperfine structure of the Rb -> hyperfine structure of Rb,

in this temperatures -> in these temperatures

charge-exchanged potential -> charge-exchange potential

transfer the atoms ... 25cm to the chamber -> transfer the atoms ... over 25cm to the chamber
radials and axial mode -> radial and axial modes
reference 11: PhysicalCetina Review Letters -> Physical Review Letters
+ many missing spaces in the references, e.g. ionatom

End of report

Reviewer #2 (Remarks to the Author):

The paper's main claim is to show in cold ion-neutral collisions that the spin or hyperfine state of an ensemble of neutral (Rb) atoms can be used to control the final spin projection of an imbedded ion (Sr^+). In addition the paper demonstrates that reactions such as charge-exchange collisions (with Sr^+) can be controlled by the hyperfine state of the target ground state atoms. What is really novel in an ultracold collision is that when the Rb atoms are pumped into the $(F, M_F) = (1, -1)$ or $(1, 0)$ initial state and using a single Sr^+ ion, the final spin projection of the ion is controlled by whether or not a spin-exchange collision is allowed from the initial state. So far as this reviewer knows, this is the first demonstration of spin-exchange control of an ion spin in an ultracold collision using a hybrid ion-neutral trap. The paper also shows control of ultracold charge-exchange collisions by the target atom hyperfine state, but this has been done before (see Ref. [5]).

The research also makes adroit use of the fact that radiative charge exchange to the ground state of the $(\text{RbSr})^+$ molecule (in a singlet Sigma state) requires a singlet Sigma entrance channel in sorting out the processes that can take place. Suppression of charge-exchange collisions is shown to occur when the Rb target state is the $F=2$ rather than the $F=1$ hyperfine state. The final spin polarization of the Sr^+ ion is found to agree with simple rate-equation theories. The work is of high quality and deserves publication in my opinion.

We thank the referees for their careful reading of our manuscript and useful remarks.

We agree with most of their suggestions and therefore implemented them in the manuscript. See below our point-by-point reply.

Reviewers' comments:

Reviewer #1 (Remarks to the Author):

The authors investigate inelastic collisions between a spin polarized Sr⁺ ion and spin polarized ⁸⁷Rb atoms. After a given interaction time they investigate whether a charge exchange has taken place or whether the spin state of the Sr⁺ has flipped. In general, depending on the initial spin states of ion and atoms, the final product populations vary. As a first important feature the authors observe collisional spin pumping--- i.e. collisions with the polarized atoms can polarize the ion spin. In contrast to the work of the Köhl group spin relaxation plays a minor role here and collisions are dominated by spin exchange processes. Another important result is that the data suggest that two-body charge exchange is only taking place within the electronic singlet channel. This is expected if the charge exchange process involves a radiative decay to the electronic ground state.

The presented results are important for the new field of cold atom-ion collisions. The manuscript is written in a clear and intelligible manner. There are, however, a number of typos and some of the English should be checked again.

I recommend the manuscript for publication in Nature Communications after the following issues have been dealt with:

1) On page 2 the authors claim that no spin control of collisional properties was demonstrated to date with cold atom-ion collisions. In view of ref 24 this statement needs to be toned down, in my opinion.

In the article of Ratschbacher et al. authors study the evolution of hyperfine and Zeeman spin states of the Yb⁺ ion immersed into a cloud of Rb atoms. They report a spin-relaxation to be two times faster than spin-exchange. The ion relaxes into a nearly fully-mixed state after few collisions. The steady-state spin polarization of the ion can be only mildly tuned (between 0.4-0.6) by controlling the atomic polarization (see [24] Fig. 2c). Due to the presence of strong spin-nonconserving mechanism the spin of the ion can be neither controlled nor preserved during collisions, and therefore no control of collisional properties was demonstrated.

2) page 1: "... can only be reached by initially overlapping a singlet manifold..." This sentence is not clear.

We agree with the referee that this sentence could have been phrased better. This sentence is supposed to explain that radiative charge-exchange can only proceed from a singlet state and not from the triplet state. We have rewritten the sentence such that it is more clear.

3) Equation (1). The equation seems wrong. The tensor product should be a scalar product. Yes. We thank the reviewer for finding this typo.

4) page 2: “Spin-orbit interactions mixes... and leads to spin relaxation.” The singlet-triplet mixing is not what is really behind spin relaxation.

The spin-orbit interaction mixes the states with $l=0$ and states with $l=1$. We modified the sentence.

5) “Spin-projection is ... no longer conserved”. This statement is somewhat sloppy and not clear enough.

What we mean here is that during a collision the singlet and triplet states are no longer eigenstates. We modified the sentence such that it is more clear.

6) “... time of 500ms, during which 10’s of Langevin collisions occurred,” shouldn’t it be hundreds of collisions?

Yes. We corrected that.

7) page 6: What is the spin relaxation rate for $F = 2$?

As shown in the supplementary material Fig. S1, the collisions with atoms at $F=2$ lead to heating of the ion to temperatures above 10 mK due to occasional Rb hyperfine energy (330 mK) release. At these temperatures, the ion is outside the Lamb-Dicke regime and therefore spin detection through 674nm S->D electron shelving becomes unreliable. We, therefore, did not measure the spin relaxation rate for $F=2$ directly.

8) page 6: Defining ultracold collisions to occur when the mK regime is reached, is kind of arbitrary. A better definition for the realm of ultracold collisions is connected to reaching the s-wave regime.

We agree that the statement about an ultracold regime that we used was not very exact. This paragraph talks about the hyperfine barrier in Rb atoms. It explains that we worked in an energy regime (few mK) which is two orders of magnitude smaller than hyperfine barrier (330 mK) and therefore transitions from $F=1$ to $F=2$ are energetically forbidden. We rephrased the paragraph and removed the claim about the ultracold regime.

9) page 10: The following sentence is confusing as it does not clearly explain how the experiment is done: “We overlapped... for a duration of 10^6 Langevin collisions and have recorded 104 ... events”. How many runs were done? How long (i.e. how many Langevin collisions) was every run?

We agree that this paragraph was written too concisely. We now give a more detailed description of the experimental procedure.

Typos:

gasses -> gases;

hyperfine structure of the Rb -> hyperfine structure of Rb,

in this temperatures -> in these temperatures

charge-exchanged potential -> charge-exchange potential

transfer the atoms ... 25cm to the chamber -> transfer the atoms ... over 25cm to the chamber

radials and axial mode -> radial and axial modes
reference 11: PhysicalCetina Review Letters -> Physical Review Letters
+ many missing spaces in the references, e.g. ionatom

End of report

We corrected all typos that reviewers pointed out.

Reviewer #2 (Remarks to the Author):

The paper's main claim is to show in cold ion-neutral collisions that the spin or hyperfine state of an ensemble of neutral (Rb) atoms can be used to control the final spin projection of an imbedded ion (Sr⁺). In addition the paper demonstrates that reactions such as charge-exchange collisions (with Sr⁺) can be controlled by the hyperfine state of the target ground state atoms. What is really novel in an ultracold collision is that when the Rb atoms are pumped into the (F, M_F) = (1,-1) or (1,0) initial state and using a single Sr⁺ ion, the final spin projection of the ion is controlled by whether or not a spin-exchange collision is allowed from the initial state. So far as this reviewer knows, this is the first demonstration of spin-exchange control of an ion spin in an ultracold collision using a hybrid ion-neutral trap. The paper also shows control of ultracold charge-exchange collisions by the target atom hyperfine state, but this has been done before (see Ref. [5]).

The research also makes adroit use of the fact that radiative charge exchange to the ground state of the (RbSr)⁺ molecule (in a singlet Sigma state) requires a singlet Sigma entrance channel in sorting out the processes that can take place. Suppression of charge-exchange collisions is shown to occur when the Rb target state is the F=2 rather than the F=1 hyperfine state. The final spin polarization of the Sr⁺ ion is found to agree with simple rate-equation theories. The work is of high quality and deserves publication in my opinion.

We appreciate Reviewer #2 positive feedback.

We also shortened the abstract such that it is below 150 words and modified the title to: Spin controlled atom-ion chemistry.

REVIEWERS' COMMENTS:

Reviewer #1 (Remarks to the Author):

The authors have responded satisfactorily to the issues raised and have correspondingly changed their manuscript.

The manuscript is now ready for publication in Nature Communications.